# When a Synonymous Variant Is Nonsynonymous

**DOI:** 10.3390/genes13081485

**Published:** 2022-08-19

**Authors:** Mauno Vihinen

**Affiliations:** Department of Experimental Medical Science, BMC B13, Lund University, SE-22 184 Lund, Sweden; mauno.vihinenl@med.lu.se

**Keywords:** variation interpretation, synonymous variation, mutation, terminology, variation naming, annotation

## Abstract

Term synonymous variation is widely used, but frequently in a wrong or misleading meaning and context. Twenty three point eight % of possible nucleotide substitution types in the universal genetic code are for synonymous amino acid changes, but when these variants have a phenotype and functional effect, they are very seldom synonymous. Such variants may manifest changes at DNA, RNA and/or protein levels. Large numbers of variations are erroneously annotated as synonymous, which causes problems e.g., in clinical genetics and diagnosis of diseases. To facilitate precise communication, novel systematics and nomenclature are introduced for variants that when looking only at the genetic code seem like synonymous, but which have phenotypes. A new term, unsense variant is defined as a substitution in the mRNA coding region that affects gene expression and protein production without introducing a stop codon in the variation site. Such variants are common and need to be correctly annotated. Proper naming and annotation are important also to increase awareness of these variants and their consequences.

## 1. Introduction

Terminology in genetics is very important for precise transfer of information. Still, it is common to use confusing and even wrong terms, e.g., for variant types [1]. Variation Ontology (VariO) facilitates systematic naming and annotation of variants and their effects, mechanisms and other characteristics at DNA, RNA and protein levels [2]. Despite the availability of VariO and other systematics and recommendations, such as HGVS nomenclature [3], variation naming is often confusing, misinterpreted and sometimes outright misleading. Some problems and solutions in variation naming have been discussed previously [1], here the focus is on synonymous variants.

Description of RNA variations can be complicated as there are numerous types [4], see Figure 1. Starting with the genetics textbooks, single nucleotide variants (SNVs) in mRNA coding region are wrongly classified only to three classes: missense, nonsense and synonymous (also called silent). These terms refer to variant descriptions in relation to codon triplet code in protein translation. Synonymous means an RNA variant that does not change amino acid in the coded protein. The 20 types of amino acids that appear in proteins are coded by 61 codons (with 3 stop codons there are totally 64 codons), 1 to 6 different codons per amino acid. When a substitution is by another codon for the same amino acid it is synonymous. Several variants are *bona fide* synonymous; however, when such a variant has a phenotype and functional effect, it is very seldom synonymous. Mechanisms of these variants have been previously described in [5,6,7]. Here, systematics is introduced for synonymous variants and those that may look like synonymous, but without being such, to facilitate correct annotation of variations. Wrong annotations of variants that look like synonymous, which in fact affect the coded protein, are usually ignored in variation interpretation and therefore genetic diagnosis is not made even when there is a genetic cause. It is important for users of genetic information to be aware of these kinds of cases and to properly consider such instances in genetic studies.

## 2. Consequences of Synonymous Variants

Synonymous variations are common. Methionine and tryptophan are the only amino acids that are coded by a single codon, thus all other amino acids have at least two codons. The genetic code is composed such that the synonymous codons are usually interchangeable by a single substitution. 141 (24.5%) out of the 576 possible substitutions due to SNVs (64 × 9, nine variants per codon for 64 codons) are synonymous. Four of the substitutions are between stop codons, so 137 (23.8%) are between coding codons. Only in 8 (1.4%) of the synonymous codons the difference is outside the third codon position that is called the wobble position. All the 8 differences are in the first codon position. For 7 (43.8%) of the 16 codon groups in the genetic code, the third position bases are all interchangeable, i.e., synonymous.

A variant may seem like synonymous, but can have a fundamental effect on DNA, RNA and/or protein level. Therefore, descriptions of variants have to look beyond the genetic code. Variants that are erroneously called synonymous have a phenotype and affect the translated protein.

Synonymous refers to RNA and protein levels; however, these variants can have a DNA level effect on transcription factor (TF) binding (Figure 2). Both in protein coding and non-coding regions such alterations can impair TF binding to the DNA and affect transcription and protein production. Many exonic variants, including synonymous alterations, affect transcription factor occupancy [8]. About 15% of human codons were predicted to be dual-use-codons (duons) and act also as TF binding sites.

Putative synonymous variants can have various effects on RNA level [4] (Figure 2). Most frequently such variants with an effect impair splicing, either by altering splicing site [9] or splicing regulatory motif, such as exonic splicing enhancer (ESE) [10] or exonic splicing silencer (ESS) [11]. As an example, C to T transition at position 6 in exon 7 of the *SMN2* gene creates an ESS and impairs splicing leading to spinal muscular atrophy [12,13].

The consequences of these variants include changes to alternative splicing, exon skipping, exon inclusion and other forms of aberrant splicing, which frequently lead to premature stop codon and mRNA degradation by mRNA quality control machineries, especially by nonsense mediated decay (NMD) [14]. A variant in a non-existent RNA is not silent, since it does not exist at all and therefore there is no protein either. This kind of variants are selected in cancers [15]. NMD escape can occur in certain exons and cases [16]. If the length of deleted/inserted mRNA segment is divisible by three it can possibly retain the coding frame and not lead to premature termination and mRNA degradation. Changes in the last exon and towards the end of the penultimate exon are more tolerated or at least can code for a protein as NMD does not detect and destroy the messenger RNAs.

A variant can change mRNA structure and stability and thereby affect the abundance of the produced protein [17] (Figure 2). Single stranded mRNA molecules form Watson-Crick base pairs and fold into complex three-dimensional structures. These structures increase intracellular stability of otherwise vulnerable mRNA molecules. Synonymous substitution can alter base pair formation and have a major impact on the structure and stability of mRNA and consequently on gene expression and protein abundance [17].

Variants may alter also exonic miRNA binding sites and impair gene expression regulation. miRNA binding sites are common also in coding regions of transcripts [18]. A C > T transition in codon for leucine 105 in *IRGM* gene for immunity related GTPase M destroys binding site for miR-19 [19]. The loss of the binding site due to an apparently synonymous variant impairs the control of intracellular replication of invasive *Escherichia coli* bacteria by autophagy in Crohn’s disease, a type of inflammatory bowel disease.

At protein level, a synonymous variant can affect protein translation due to codon usage bias and limitations in the aminoacyl-tRNA pool [20]. Alteration by a rare codon, although synonymous, can affect translation speed [21] and co-translational protein folding [22] causing conformational change [23] and even protein misfolding and missing activity [24], increased specific activity [25], or other altered properties [26]. Numerous additional protein level effects are possible, including changes to post translational modifications, with effects on regulation and activity [9]. As an example, synonymous variation in *TP53* codon 22 for leucine prevents phosphorylation of the coded protein and affects its stability [27].

In summary, variants that look at codon level as synonymous can have numerous consequences. Therefore, variants should be called as synonymous only when supported by experimental evidence, similar to variants affecting function should be supported by evidence [28]. Many of the wrongly as synonymous annotated variants affect protein abundance either due to effect on expression or because of mRNA degradation by quality control systems.

## 3. Problematic Databases and Annotations

The use of term synonymous in wrong contexts is widespread. The HGVS nomenclature for variants [3] does not provide description for these variants and therefore they are annotated as synonymous. Annotation after variant calling is a crucial step in sequencing projects. The available annotation tools, including ANNOVAR [29] and SnpEff [30], have just one category for synonymous variants. Therefore, in variation interpretation these variants are usually ignored and therefore disease diagnosis may be prevented or substantially delayed.

Variation naming tools Mutalyzer [31], VariantValidator [32] and hgvs Python package [33] and tools like Variant Effect Predictor (VEP) [34] categorically and wrongly classify variants as synonymous based on codon information as they follow the HGVS recommendations.

Certain databases lump together different types of variants as synonymous, examples include DMSN [35] and SynMICdb [36]. Database of Deleterious Synonymous Mutations (dbDSM) [37] aims to provide mechanism-based classification for synonymous variants. Even dbDSM contains wrong and misleading information, large portion of the variants are markers used in genome-wide association studies (GWASs), and as such are hardly ever disease-causing. Data in locus specific databases, most often in LOVD system [38], are of variable quality in this regard. sSNV for synonymous single nucleotide variant is widely misused in them being without knowledge of effect(s).

Analyses of putative synonymous variants without considering the actual mechanisms, e.g., [36,39], are of limited value as different aspects are relevant for different mechanisms. Another problematic application area is in variation interpretation. Several predictors claim to predict effects of synonymous variants, although they actually are trained to detect non-synonymous unsense variants, examples include TraP [40], usDSM [41] and DDIG-IN [42]. The training data for usDSM contains also GWAS markers, which are irrelevant for this prediction task.

## 4. Systematic Nomenclature

It is evident that improved systematics is needed for synonymous and wrongly as synonymous annotated variants. The new systematics in Figure 2 depicts the effects and mechanisms of synonymous and putative synonymous variants at DNA, RNA and protein levels. The variants and effects on DNA and protein level can be described with existing systems as VariO; however, there has not been terminology for the RNA level effects. One could claim that the variants discussed here can be grouped under the umbrella term nonsynonymous variants. This is naturally true, but it does not help naming the variant categories in a specific manner. For variants to be considered and taken into account, they need to be annotated and thus have a specific name.

There are hundreds of known disease-related variations in dbDSM [37], many more will be found once the variants are properly investigated. Experimental studies show that a substantial part of exonic substitutions, both synonymous and nonsynonymous, affect splicing.

Minigenes include an exon or exons and flanking control regions. They are used to investigate splicing effects on gene expression. Combined with massively parallel reporter assays (also called saturation mutagenesis), minigenes facilitate experimental investigation of large numbers of splicing effects. Several studies have shown that a substantial number of exonic substitutions affect splicing. 23% (32/138) of the possible synonymous variants in the exon 7 of *SMN1* gene decreased exon inclusion [43]. In another study, 6.3% of 725 de novo coding region variants identified in autism spectrum disorder families disrupted splicing [44] and included also variants annotated as synonymous.

To facilitate precise description of these variants at RNA level, a new concept—unsense (meaning lack or absence of sense, senselessness, or nonsense) variant is introduced. The new term along with the other RNA substitution types are defined in Figure 3. The definitions are from VariO.

Unsense as a variant type is in line with nonsense and missense, the sense is lost in unsense variants as the expression of the coded protein is affected. Although unsense variants often lead to missing protein, they are different from nonsense variants that are due to introduced stop codons. Unsense variants impair splicing, splicing regulation or gene expression due to an exonic substitution.

The most common unsense variants impair splicing, either by altering or generating an exonic splice site [45] or modifying an exonic splicing regulator, ESE or ESS [46]. Most of these variants introduce a premature stop codon, are recognized and degraded by NMD and are thus of type missing RNA. Even when the transcript is not degraded, the sense of the protein is lost due to insertion, deletion or frameshift alteration. The third mechanism for unsense variations alters exonic regulatory miRNA binding sites and impairs miRNA regulation of gene expression [47]. Additional mechanisms of unsense variations may by detected when experimental studies to reveal variation mechanisms are performed.

## 5. Discussion

A variant may at codon level look as synonymous but should be called as such only when supported by experimental evidence, similar to variants affecting function should be supported by evidence to avoid confusion and wrong conclusions [28]. The full description of synonymous, or any variation, should include in addition to the variation type information also effects and consequences in all the relevant levels, thus often also RNA and protein alterations and effects.

If a variant is considered as synonymous without experimental evidence, there is a risk of wrong description. Variation annotation pipelines outcomes need to be updated and use appropriate terminology for accurate variant naming. They are dependent on the terminology, which should be updated in the first place. Studies that pool all apparently synonymous variants together without further analysis and consideration likely provide at least somewhat or partially wrong or biased insight. Curators of variant databases should be cautious with term synonymous and reserve it only for verified instances.

It is also problematic that systematics, such as Sequence Ontology (SO) annotations [48] and variant naming tools [31,32,33], do not allow proper annotations. If a variant is suspected to be synonymous, it should be presented as such and not stated as a fact.

To implement the new terminology, HGVS nomenclature should be updated and then implemented to annotation and variant naming tools. It will be possible to write scripts to edit also existing information e.g., in variation databases. This would require marking many current annotations of synonymous variants as predictions due to lack of experimental evidence. It will allow users to see variants which would need experimental investigation. Verified synonymous and unsense cases need to be properly annotated and the new term makes it possible.

It is important to notice that all apparently synonymous variants are not equal, and not even synonymous.

## Figures and Tables

**Figure 1 genes-13-01485-f001:**
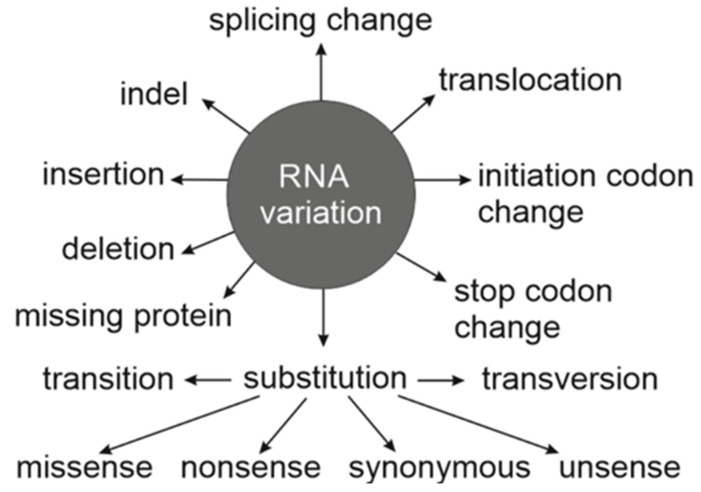
Types of RNA variations as described in VariO. Substitutions are divided into six categories. Transitions and transversions describe the chemical change of the base group in a nucleotide. Missense, nonsense, synonymous and the new term unsense describe the outcomes of the alterations in relation to translation, splicing, regulation and function.

**Figure 2 genes-13-01485-f002:**
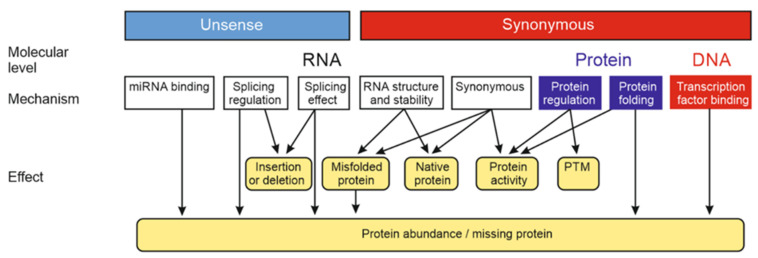
Novel classification of unsense and synonymous variations, molecular levels where they have functions, their mechanisms, as well as effects. Synonymous variants can affect DNA (red boxes), RNA (white) and protein levels (blue), whereas unsense variants are at RNA level and affect splicing, regulation or binding. PTM, post translational modification.

**Figure 3 genes-13-01485-f003:**
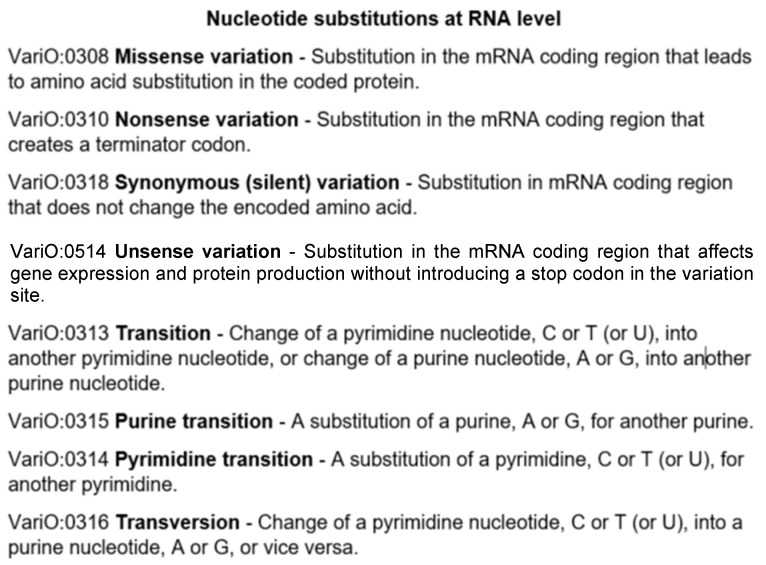
Types of RNA substitutions and VariO codes, and their definitions.

## Data Availability

Not applicable.

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
