# Peer review of "When a Synonymous Variant Is Nonsynonymous"

_genes, 2022, doi:10.3390/genes13081485_

Round 1

Reviewer 1 Report

Major comments:

The article focuses on synonymous variants consequences, whith rich references and accurate interpretations.

As described in the present article and literature, the term “synonymous” refers on the effect on the translated amino acid, and do not imply that a synonymous variants has no biological effect.

Although an updated terminology clarifying the fact that a synonymous variant might, or not, be deleterious and disease causing, the word “unsense” itself might be easily confounded with nonsense and it’s meaning remains unclear given that all synonymous variants affecting protein function or abundance are not all considered as “unsense” according to the manuscript (Figure 2). Would it be of interest to rather propose categories inside the “synonymous” class ?

A statement issued by consortium of professionnals of genetics and molecular biology would be of interest there, as it is true that synonymous variants effects are often underestimated in clinical practice and research procedures.

As a consequence:

13: I would remove this sentence (“A new term,…”)

Figure 2: my suggestion would be to remove the blue and red boxes, and show there the different effects of a synonymous variants. In this figure, the terminology in the small yellow boxes seems unclear (there is no arrow from protein folding to misfoldes protein for example). This figure would benefit from further clarifications, but will be, after a few improvements, a helpful reference for the community.

The Discussion section is pretty short and a possibility would be to discuss there the options for terminology change.

Minor comments:

7: “23.8%” please write in whole letters given that it’s the beginning of the sentence

52: the term missense has also been debated, to use nonsynonymous instead

68: “Variant that are erroneously called synonymous -can lead to a phenotype-(have a phenotyp and affect the translated protein.” An absence of phenotypic change is also a possibility with a synonymous variant.

203: “Variation annotation pipelines are the main sources of these 203 problems.” This is a strong statement. Please consider removing this sentence and replace by: “Variation annotation pipelines outcomes need an updated and appropriate terminology for an accurate variant effect interpretation”

Author Response

I am thankful for the thorough and expert review.

Responses to major comments:

I respectfully disagree on the suggestion to have categories for synonymous variants instead of introducing the new term unsense for two reasons. First, unsense variants are not synonymous, they affect the protein sequence, often preventing protein synthesis completely as the mRNA is degraded. This is truly the fourth category of substitutions in mRNA along with missense, nonsense and synonymous. Second, it is important to have a name for these instances e.g. to increase awareness. Things for which there are not names and defined concepts are easily forgotten and neglected. Specific definitions are provided for unsense, nonsense and the other discussed variants in Figure 3 and they are implemented to Variation Ontology.

I fully agree on the statement from a professional consortium, however, feel that it is best to bring the issue up and then allow the professionals to react. Although there are genetics societies, none of them has responsibility e.g. for correct annotations. I was given five days to submit the edits. As a member in the boards of some scientific societies, I know from experience that to get such a statement could take rather five months (or more) than five days.

Response to line 13 comment: See my comment above.

I have added discussion about how to implement the new terminology in practice.

Responses to minor comments:

Line 7: Done

Line 52: This is true. Missense is a specific type of nonsynonymous variants, and thus needed.

Line 68: Thank you for the comment. This is indicated in Figure 2.

Line 203: Thank you for the nice wording! I have made the edit, the message is still the same.

Reviewer 2 Report

The author has described a new terminology of unsense which is novel in terms of genomics, where there is an issue in understanding synonymous mutations. I think the author of this review article has tried to address this issue and suggested the terminology of unsense mutation. 

Author Response

I want to thank for the supportive comments.